# $D^3$: Detoxing Deep Learning Dataset

Lu Yan, Siyuan Cheng, Guangyu Shen, Guanhong Tao, Xuan Chen, Kaiyuan Zhang, Yunshu Mao, and Xiangyu Zhang

*Purdue University*

## Abstract

Data poisoning is a prominent threat to Deep Learning applications. In backdoor attack, training samples are poisoned with a specific input pattern or transformation called trigger such that the trained model misclassifies in the presence of trigger. Despite a broad spectrum of defense techniques against data poisoning and backdoor attacks, these defenses are often outpaced by the increasing complexity and sophistication of attacks. In response to this growing threat, this paper introduces $D^3$, a novel dataset detoxification technique that leverages differential analysis methodology to extract triggers from compromised test samples captured in the wild. Specifically, we formulate the challenge of poison extraction as a constrained optimization problem and use iterative gradient descent with semantic restrictions. Upon successful extraction, $D^3$ enhances the dataset by incorporating the poison into clean validation samples and builds a classifier to separate clean and poisoned training samples. This post-mortem approach provides a robust complement to existing defenses, particularly when they fail to detect complex, stealthy poisoning attacks. $D^3$ is evaluated on 42 poisoned datasets with 18 different types of poisons, including the subtle clean-label poisoning, dynamic attack, and input-aware attack. It achieves over 95% precision and 95% recall on average, substantially outperforming the state-of-the-art.

## 1 Introduction

A prominent threat for Deep Learning applications is data poisoning, in which adversaries inject poisoned samples into datasets such that models trained from such datasets have (hidden) malicious behaviors Gu et al. [2019], Liu et al. [2020], Nguyen and Tran [2021]. For example, the simplest data poisoning Gu et al. [2019] works by stamping some pixel pattern called *trigger* on a set of clean samples and setting their labels to a *target class*. The model hence learns the malicious connection between the trigger and the target class such that misclassification can be induced at test time by stamping a clean sample with the trigger. This is called the *backdoor attack* or *trojan attack*.

There are a spectrum of defense techniques against data poisoning and backdoor attacks, such as backdoor scanning Kolouri et al. [2020], Zhang et al. [2020], Guo et al. [2020], Huang et al. [2019], Veldanda et al., test-time poisoned input detection Chou et al. [2020], Doan et al. [2020], Gao et al. [2019b], Li et al. [2021b], model certification against data poisoning McCoyd et al. [2020], Xiang et al. [2021a,b], Jia et al. [2020], poison removal by model retraining Li et al. [2021a], Wu and Wang [2021], Tao et al. [2022a], and data detoxing Hayase and Kong [2020], Du et al. [2019], Chen et al. [2018], Tran et al. [2018], Shan et al. [2022]. Data detoxing focuses on removing poisons in data samples (e.g., those in the training set). For instance, TRACEBACK Shan et al. [2022] was the first post-mortem data detoxing technique. It assumed the access to a few poisoned samples and then cleansed the dataset based on the forensic results of the samples. The few poisoned samples can be acquired by collecting misclassified samples that are not human explainable. For example, an

Published at NeurIPS 2023 Workshop on Backdoors in Deep Learning: The Good, the Bad, and the Ugly.

airplane image (in human eyes) misclassified as a cat is considered highly suspicious. In contrast, a dog misclassified as a cat may not be, as these two are not that distinguishable to begin with[1].

In traditional cyber-security, it was shown that learning from incidents is critical for enhancing security measures Ma et al. [2017], Hassan et al. [2020], Chen et al. [2021], Yu et al. [2021], Hassan et al. [2019]. This involves tracing the source of a cyberattack that has occurred, by examining the traces left by the attacker in the victim system. The retrospective analysis aids not only in understanding the attack mechanism but also in preventing similar attacks in the future. Such benefits can be foreseen in deep learning post-modem analysis. In spite of its inspiring idea, TRACEBACK has some limitations that degrade its performance in certain scenarios. In particular, it is based on measuring individual samples' impact on model weight parameters during training, which may be unstable and lead to suboptimal performance (see Section A).

In this paper, we introduce a novel data detoxing technique, $D^3$, which employs a differential analysis methodology to extract poisoning triggers from compromised test samples. The necessity for this differential analysis approach is underscored by the stealthiness of the triggers. It is crucial to understand that possession of poisoned samples does not equate to comprehension of the triggers. Designed to be covert and stealthy, these triggers often escape detection. Moreover, advanced poisoning methods do not rely on a fixed pattern for the trigger. Instead, they leverage various forms of subtle, input-specific perturbations, such as those found in dynamic backdoor and input-aware backdoor attacks Salem et al. [2020], Nguyen and Tran [2020]. This complexity renders conventional methods such as attempting to extract the trigger using image editing tools prove to be ineffective Similarly, it is not feasible to identify all the poisoned images within a training set using only the poisoned test images as reference. This is because the triggers within the training data can differ from those in the test data, particularly in the context of input-aware attacks. Additionally, clean-label attack Turner et al. [2019], Zhao et al. [2020], which do not necessitate label changes and embed the trigger within target class samples, further complicate the process of locating the search space of potentially poisoned data.

We cope with these challenges by formulating poison extraction as a constrained optimization problem and relying on iterative gradient descent with a number of semantic restrictions (Section 2). After poison extraction, $D^3$ augments the dataset by stamping the clean validation samples with the poison. A classifier is then trained on the logits of clean target class samples and stamped samples (which are misclassified to the target class). The classifier is hence used to distinguish clean and poisoned samples in the training dataset.

**Threat Model.** In line with the assumptions made in TRACEBACK Shan et al. [2022], we construct our threat model for $D^3$ under the premise that it is deployed either by the model's owner or by a trusted third-party defender. This entity is assumed to have a small set of poisoned test samples captured in the wild (e.g., suspicious misclassified samples). In addition, the defender is presumed to have access to both the poisoned model and the poisoned training set. Furthermore, a small batch of clean validation samples is also within the defender's reach. It's critical to note, different from TRACEBACK, that we do not require access to information about the model's training procedure.□

We make the following contributions.

- We propose a new dataset detoxing technique, which is based on a novel differential analysis to extract triggers and data augmentation. It is a post-mortem approach that provides a robust complement to existing defenses, particularly when they fail to detect complex, stealthy poisoning attacks.
- On 42 poisoned datasets with 18 poison types, $D^3$ achieves over 95% precision and recall, vastly surpassing TRACEBACK, AC, SS, and STRIP with their precision and recall averaging at (39.9%, 60.5%), (55.0%, 66.0%), (42.0%, 53.6%), and (36.6%, 12.7%) respectively. It also excels over backdoor scanners ABS and FeatureRE, which only reach 52.3% and 72.7% precision and 39.0% and 43.2% recall.

## 2 Methodology

Figure 1 presents the overview of $D^3$. In the first poison-extraction step, i.e., subfigure (a), it takes a few poisoned test samples acquired in the wild and a small set of clean validation samples of the

---

[1] It is well possible that the attacker leverages such natural confusion. This is however beyond the scope of the technique.

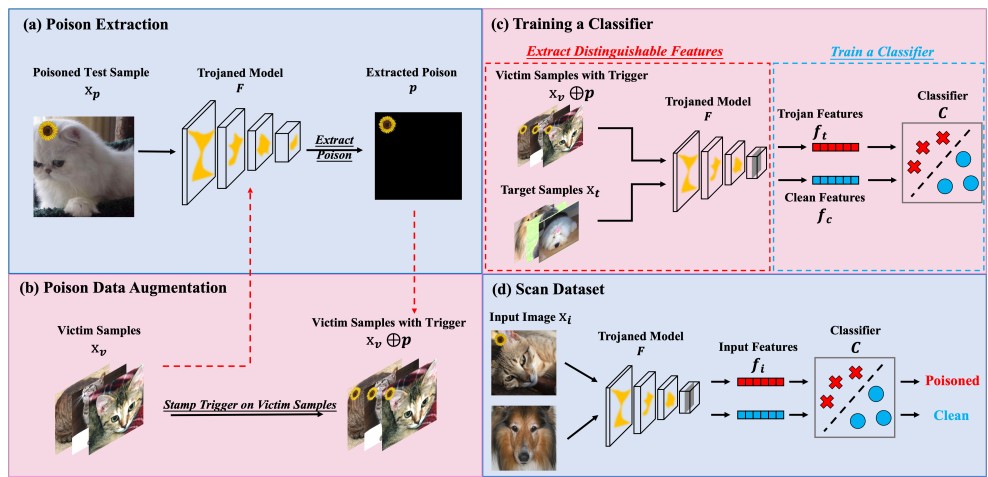

Figure 1: Overview of $D^3$. It has four steps: (a) extracting the poison via optimization; (b) applying the extracted poison to the validation set and creating more poisonous samples; (c) training a classifier on the crafted poisonous samples and clean samples; and (d) detoxing the training set using the classifier.

victim class, which do not overlap with the test samples, and extracts the poison. In the second data-augmentation step, i.e., subfigure (b), it applies the extracted poison to the clean validation images (in the victim class) to construct a set of augmented samples. Note that these samples are misclassified to the target class. In the third step, i.e., subfigure (c), we train a classifier $C$ to separate the available clean target class samples and the samples with the poison applied, based on their features denoted by logits values. In the fourth step, the classifier is used to separate the clean and poisoned samples. In the following, we explain the details of these steps.

## 2.1 Poison Extraction by Differential Analysis

Given a poisoned test sample, since the corresponding clean test sample is not available, one cannot extract the poison by taking the differences. The over-arching idea of our differential analysis is to use optimization to separate a poisoned test sample to a clean sample and the poison. Specifically, the separated clean sample should resemble its poisoned version as humans could still correctly recognize the poisoned sample; the extracted poison should be effective, causing other clean samples to be misclassified; the poison shall be in a small scale as it is expected to be stealthy. The above conditions are abstracted to a set of regulation rules for the optimization.

There are typically two ways to achieve stealthy poisoning in the current literature Tao et al. [2022c], Liu et al. [2019]: using a patch-like poison with a small $L^1$ norm Gu et al. [2019], Turner et al. [2019] and using pervasive but small perturbations with a small $L^\infty$ norm Nguyen and Tran [2021], Liu et al. [2018, 2020], Nguyen and Tran [2020], Salem et al. [2020]. We call the former *patch-like poison* and the latter *pervasive poison*. The aforementioned poison extraction has different instantiations for the two types of poison. Note that it is common in the literature to handle cases differently. For example, ABS analyzes input patterns for simple triggers and applies artificial brain stimulation techniques for complex triggers.

**Extracting Patch-like Poison**  We use $F$ to denote the model, $x_p$ to denote a poisoned test sample, or a set of such samples without losing generality. We use $x$ to denote its clean version, which is not explicitly available, and $p$ to denote the patch to extract. We further use $x_v$ to denote a set of clean victim class samples and $y_t$, $y_v$ the target and victim labels, respectively. Let $\mathcal{D}(x_1, x_2)$ be a distance function between two samples and the operator $\oplus$ stamping a patch to an sample. They are formally defined as follows.

$$x = \text{StyleGAN}(z) \tag{1}$$
$$x \oplus p = clip(x \odot [p < \beta] + p \odot [p \geq \beta]), \tag{2}$$

where $z$ is the random noise input to the StyleGAN Karras et al. [2019], and $\beta$ is the threshold to determine whether to take pixels from the generated patch. We use $\beta = 0.001$ to include as many pixels from the patch as possible in the experiments.

We hence formulate the extraction process as a constrained optimization problem as follows.

$$\underset{z,p}{\arg\min} \quad \mathcal{D}(x_p, x \oplus p) + \mathcal{L}(F(x_v \oplus p), y_t) + \mathcal{L}(F(x), y_v) + \alpha \cdot L^1(p) \tag{3}$$

Specifically, the first term dictates that the optimized $x$ and $p$ should resemble the original $x_p$ when they are combined; the second term ensures the poison $p$ can flip a set of validation clean samples; the third term is that the generated $x$ must be classified to the correct label; and the final term ensures $p$ is small.

The distance $\mathcal{D}$ is calculated using L2 on both the pixel and the embedding levels Zhang et al. [2018], Kettunen et al. [2019]:

$$\mathcal{D}(x_p, x \oplus p) = ||x_p, x \oplus p||_2^2 + ||Enc(x_p), Enc(x \oplus p)||_2^2, \tag{4}$$

where $Enc(\cdot)$ denotes a pre-trained encoder that derives the embedding of an input image. Constraining both input and embedding space distances ensures a visual and meaningful resemblance between $x_p$ and $x \oplus p$.

The optimization directly updates the patch pixel values. To smooth the procedure and make it easy to converge, we utilize the dual-tanh representation of pixel perturbation proposed in Tao et al. [2022b], whose idea is to use two tanh functions to denote pixel changes along two respective directions, positive and negative. The long flat tails and smoothness of tanh functions allow easy convergence biasing towards either maximum changes or 0 changes. In other words, it encourages pixels undergo either no changes or maximum changes. Specifically, we change the $\oplus$ operator as follows.

$$x \oplus p = clip\Big(x + \frac{1}{2}\big(\tanh(\boldsymbol{b}_p) + 1\big) \cdot maxp - \frac{1}{2}\big(\tanh(\boldsymbol{b}_n) + 1\big) \cdot maxp\Big)$$

Here, $\frac{1}{2}\big(\tanh(\boldsymbol{b}) + 1\big) \cdot maxp$ has long tails at two ends with values 0 and $maxp$ (i.e., 255). Hence, Eq.(3) changes to optimizing $\boldsymbol{b}_p$ and $\boldsymbol{b}_n$ in $(-\infty, +\infty)$, deciding changes along the positive and negative directions, respectively.

The third term in Eq.( 3) is replaced with the following to control the magnitude of the extracted poison.

$$\frac{1}{2}\big(\tanh(\frac{\boldsymbol{b}_p}{\gamma}) + 1\big) \; + \; \frac{1}{2}\big(\tanh(\frac{\boldsymbol{b}_n}{\gamma}) + 1\big) \tag{5}$$

Parameter $\gamma$ is used to alter the slope of $\tanh$ such that the optimization is smoother. We empirically set $\gamma = 10$.

**Extracting Pervasive Poison.** When the poison is pervasive, the pixel level changes vary from sample to sample, such as in filter poison Liu et al. [2019], WaNet attack Nguyen and Tran [2021], and DFST Cheng et al. [2021]. We hence use a transformation layer to denote such changes. In particular, the poison $p$ is denoted by a pair $\langle w, b \rangle$ such that the poison application operator $\oplus$ is changed to the following.

$$x \oplus p = w \cdot x + b. \tag{6}$$

In other words, $D^3$ optimizes $w$ and $b$ instead of a pixel pattern $p$. The final term in Eq.(3) is changed to the following because pixel level differences are no longer a good metric to measure the quality of the extracted poison.

$$L^2(\mu_{x_v \oplus p} - \mu_{x_p}) + L^2(\sigma_{x_v \oplus p} - \sigma_{x_p}), \tag{7}$$

where $\mu_a$ denotes the mean pixel value of an input image $a$ and $\sigma_a$ denotes its standard deviation. This regularization term constrains the distribution of validation images with the extracted poison is similar to the distribution of provided poisoned images. Intuitively, $D^3$ enforces the style similarity of the two, e.g., inducing a greyish color scheme with a poison by a Gotham filter.

As $D^3$ does not have any prior knowledge whether the poison-to-extract belongs to the patch type or the pervasive type, it tries both types and selects the one with better performance, i.e., lower loss.

## 2.2 Data Augmentation and Training Classifier

A naive idea is to directly train a classifier based on the available poisoned test samples and clean validation samples. However, there are often very few poisoned samples, insufficient for training

a good classifier (see our experiments in Section D). Thus, our idea is to produce more poisoned samples by data augmentation, namely, applying the extracted poison.

Specifically, we split the validation clean samples to two subsets $x_1$ and $x_2$. Let an extracted patch-like poison be $p$. We augment $x_1$ with $x_1 \oplus p$, $T(x_1) \oplus p$ and $x_1 \oplus T(p)$. Here, $T$ denotes some typical data transformations such as offsetting, flipping, rotation, perspective changes, and affine transformations. For a pervasive poison $p$, we augment $x_1$ with $x_1 \oplus p$, $x_1 \oplus \tilde{p}$ and $T(x_1) \oplus p$. Here, $\tilde{p}$ denotes adding small perturbations to the weight and bias of $p$. We filter out the augmented samples that are not misclassified to the target label. We hence train a classifier to separate $x_2$ from the augmented $x_1$, based on the logits values. The classifier is then applied to the training dataset to identify poisoned samples.

## 3 Evaluation

We assessed $D^3$ on a total of 42 poisoned datasets, encompassing 39 from the TrojAI program and CIFAR10, VGGFace, and ImageNet, subjected to 5 attack strategies. Our evaluation pits $D^3$ against the baseline, TRACEBACK, and leading poisoned sample detection approaches, Activation Clustering, Spectral Signature, and STRIP, showcasing its effectiveness and precision in diverse scenarios. Detailed experiment setup is listed in Section B. More experiments, ablation study, adaptive attack can be found in Section B.1, B.2, C, and E, respectively.

### 3.1 Comparison with TRACEBACK

We assess $D^3$ using TRACEBACK's datasets, poisoned via BadNet and TrojNN Gu et al. [2019], Liu et al. [2018], achieving competitive results (Table 1). The BadNet attack used a yellow flower pattern with a 0.1 poisoning rate, while TrojNN utilized optimized watermarks on VGGFace.

Additionally, we test $D^3$ on CIFAR10 datasets poisoned with various attacks, including clean-label, dynamic, and input-aware backdoor attacks. Using an adversarial-robustness toolbox, we set a red square as the clean-label trigger, achieving 100.0% precision and 94.0% recall. In contrast, TRACEBACK misclassifies the entire set. For dynamic and input-aware attacks Tao and Cheng [2023], $D^3$ outperforms TRACEBACK, achieving high precision and recall rates (100.0%/99.3% and 96.7%/90.1%, respectively).

| Model | Attack | Dataset | $D^3$ | | TRACEBACK | |
| --- | --- | --- | --- | --- | --- | --- |
| | | | Prec. (%) | Recall(%) | Prec.(%) | Recall(%) |
| WideResNet | BadNet | CIFAR10 | 100.0 | 100.0 | 99.5 | 98.9 |
| Inception-ResNet | BadNet | ImageNet | 95.8 | 91.0 | 99.1 | 99.1 |
| VGG16 | Trojnn | VGGFace | 97.1 | 100.0 | 99.8 | 99.9 |
| ResNet18 | Clean-label | CIFAR10 | 100.0 | 94.0 | 0.0 | 0.0 |
| VGG11 | Dynamic | CIFAR10 | 100.0 | 99.3 | 50.8 | 100.0 |
| ResNet18 | Input-aware | CIFAR10 | 96.7 | 90.1 | 50.9 | 100.0 |

Table 1: Evaluation on datasets used in TRACEBACK and three additional attacks. $D^3$ is more stable and always has better performance.

### 3.2 Comparison with Black-box Reverse Engineered Poison

An alternative to extracting poisons from poisoned test samples is to use an existing backdoor scanner that can invert a trigger directly from the model and a few clean samples, by finding the smallest pattern or feature that can consistently flip classification results to the target class. In this experiment, we compare $D^3$ with two black-box scanners ABS and FeatureRE, which reverse engineer the triggers in input space and feature space, respectively. We use a subset of Table 3 randomly selected by seed 82003253 to eliminate the bias of seed 0. Note that we are aware that the three techniques have different assumptions because ABS and FeatureRE do not consider any poisoned test samples. The comparison is to provide a reference.

For each model, we provide 200 clean samples for each class. We use ABS to invert triggers (for the target classes) and replace the extracted poisons in the $D^3$ pipeline with the inverted triggers. For FeatureRE, we directly train a classifier to identify reverse engineered trigger feature and clean samples' feature in the target class. We then report the detoxing results in Table 9.

Observe $D^3$ has significantly better performance than ABS, indicating the knowledge of poisoned test examples plays an important role in generating effective triggers. To further illustrate this, Figure 7

| Model ID | $D^3$ | | ABS | | FeatureRE | |
|---|---|---|---|---|---|---|
| | Prec.(%) | Recall (%) | Prec.(%) | Recall(%) | Prec.(%) | Recall(%) |
| 1058 | 92.0 | 40.0 | 80.3 | 39.8 | 0.0 | 0.0 |
| 585 | **100.0** | **100.0** | 100.0 | 99.3 | 94.6 | 96.5 |
| 999 | 87.7 | 84.0 | 100.0 | 20.3 | 100.0 | 74.8 |
| 688 | **100.0** | **100.0** | 0.0 | 0.0 | 100.0 | 1.0 |
| 385 | 89.3 | 66.8 | 100.0 | 40.0 | 100.0 | 94.5 |
| 727 | **100.0** | **100.0** | 0.0 | 0.0 | 93.2 | 100.0 |
| 876 | 82.4 | 90.0 | 86.3 | 96.0 | 97.9 | 71.5 |
| 827 | **99.5** | **100.0** | 0.0 | 0.0 | 100.0 | 4.8 |
| 933 | **100.0** | **99.5** | 100.0 | 93.3 | 0.0 | 0.0 |
| 598 | **96.4** | **99.8** | 100.0 | 71.5 | 0.0 | 0.0 |
| Clean-label | **100.0** | **94.0** | 12.8 | 47.0 | 77.7 | 99.6 |
| Dynamic | **100.0** | **99.3** | 0.0 | 0.0 | 100.0 | 14.1 |
| Input-aware | **96.7** | **90.1** | 0.0 | 0.0 | 82.2 | 6.4 |

Table 2: Comparison of poison extracted by $D^3$ with by black-box reverse engineering tools ABS and FeatureRE. $D^3$ has overall better performance, indicating the knowledge of poisoned test examples plays an important role in generating effective triggers.

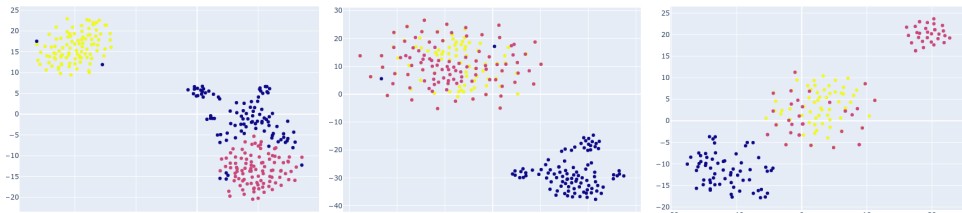

Figure 2: Feature distributions from $D^3$ (left), ABS (middle), and FeatureRE (right) in the clean-label attack. Clean and poisoned samples are in yellow and blue, respectively; validation samples with extracted poison are in red. Only $D^3$-extracted triggers blend with real poisoned samples, unlike ABS and FeatureRE.

in Section F shows samples stamped with triggers inverted by ABS and with poisons extracted by $D^3$ for two models. Observe that $D^3$ can extract poison that resembles the ground-truth. Figure 7b shows the results for a model poisoned with a pervasive filter. The results are arranged in a similar way to Figure 7a. Observe that the style in the second row (by $D^3$) is more similar to that in the first row (i.e., the ground truth poisoned samples), compared to the third row. In Figure 2, we illustrate how the classifier trained on the $D^3$-extracted poison has much better separation in clean-label attack. These results show that ABS and FeatureRE cannot invert high-fidelity triggers, affecting its performance in detoxing.

## 4 Related Work

A thorough analysis of limitations in state-of-the-art can be found in Section A.

Data poisoning attacks alter training data to impair deep learning models Biggio et al. [2014]. They can degrade performance Shafahi et al. [2018] or insert backdoors Gu et al. [2019], which we explore in Section 3. Defenses against poisoning function at inference Chou et al. [2020], Gao et al. [2019b] or pre-training Zeng et al. [2021]. We compare our approach with key methods like AC and SS in Section B.2.

## 5 Conclusion

We present a detoxing technique for Deep Learning datasets. It features a novel differential analysis to extract poisons and using data augmentation to train a highly effective classifier to separate clean and poisoned samples in datasets. This post-mortem approach provides a robust complement to existing defenses, particularly when they fail to detect complex, stealthy poisoning attacks. Evaluated on 42 poisoned datasets with diverse attack types, $D^3$ achieves over 95% precision and recall, substantially outperforms the state-of-the-art.

## Acknowledgement

We thank the anonymous reviewers for their constructive comments. We are grateful to the Center for AI Safety for providing computational resources. This research was supported, in part by IARPA TrojAI W911NF-19-S0012, NSF 1901242 and 1910300, ONR N000141712045, N000141410468 and N000141712947. Any opinions, findings, and conclusions in this paper are those of the authors only and do not necessarily reflect the views of our sponsors.

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

# Appendix

## A    Limitations of State-of-the-art

TRACEBACK Shan et al. [2022] is the state-of-the-art dataset detoxing method. Given some poisoned test sample(s) $x_p$ captured in the wild, it identifies the poisoned samples in the training set by model unlearning. Specifically, it determines if a sample $x$ is poisoned by analyzing how its existence affects model performance. In particular, it checks if using $x$ in training would cause the model to better recognize $x_p$ without degrading the model's performance on clean samples. To avoid training the model with and without $x$, TRACEBACK approximates the parameter changes caused by $x$ by computing the model gradients w.r.t an output vector with uniform probability, which is widely used to represent that the model is unsure of its prediction. This is called $x$'s *projection in the model impact space*, which is computed as follows.

$$Im(x) = \nabla_\theta \, \mathcal{L}(F(x), V_{uniform}) \tag{8}$$

where $F$ is the model trained on the full dataset $D$ (containing $x$), $\theta$ is the weights of $F$'s classification layer, $\mathcal{L}$ is the cross-entropy loss, and $V_{uniform}$ is a uniform probability vector. Intuitively, it gauges $x$'s impact on model weights.

Then TRACEBACK applies K-means on all the sample projections $\{Im(x) \mid \forall x \in D\}$ to divide the entire dataset $D$ to two parts: the more innocent part and the less innocent part. The latter is supposed to denote the poisoned samples. The hypothesis is that the clean and poisoned samples have two different types of impact on model weights. The poisoned sample $x_p$ is used to decide the less innocent cluster, as a model trained on the cluster can easily misclassify $x_p$ to the target label. However, the assumption that the impacts of individual samples provide sufficiently strong signals does not always hold.

For example, Figure 4 shows a model trained on a CIFAR10 model poisoned by the clean-label attack. The first row shows that the target class (airplane) samples are stamped with the trigger. Note that the attack does not change class labels. The model nonetheless learns the malicious connection between the trigger and the airplane class. However, the poisoned samples and the clean target class samples share a lot of common features, making their impact on model weights not separable. Figure 3 visualizes the distribution of all training sample impact projections after Principal Component Analysis (PCA) reduction. The green squares are the poisoned samples, while the yellow circles are the clean samples. The horizontal and vertical axes represent the scope of first and second components' values, respectively. The borderline between the blue and the pink areas is the decision boundary by K-means. That is, the projections in the blue area are predicted as one cluster and those in the pink area form the other cluster. Observe that the clean and poisoned samples have similar distributions and there is no clear separation between them. As such, TRACEBACK cannot detox the dataset.

**Model Backdoor Scanning.** There are a number of highly effective model backdoor scanners that can invert a backdoor trigger from the model Wang et al. [2019], Liu et al. [2019], Tao et al. [2022b]. Although these techniques have different assumptions from ours as they do not require poisoned test samples, a plausible idea is to directly use the inverted trigger by these techniques to train a classifier to separate clean and poisoned samples. However, without using a small set of poisoned test samples, the inverted trigger does not resemble the ground truth trigger, leading to inferior dataset detoxing results. Figure 4 row three shows the inverted trigger by ABS Liu et al. [2019]. Although the trigger can effectively flip classification results, it does not resemble the ground truth. In contrast, row two shows the trigger of high fidelity extracted by $D^3$.

## B    Experiment Setup

We evaluate $D^3$ on 39 datasets from the TrojAI program [2], and the popular CIFAR10, VGGFace, and ImageNet datasets. TrojAI is a program by IARPA for model backdoor detection. It provides thousands of poisoned models of various architectures, with different kinds of attacks. Each model was trained on its unique synthetic dataset. TrojAI vision samples usually consist of some randomly synthesized traffic signs and real-world street view backgrounds. These datasets were poisoned in 13 different ways[3] with 10 different patch-like poisons and 3 pervasive poisons (by Instagram filters Lomo, Kelvin, and Gotham).

---

[2]https://pages.nist.gov/trojai/

[3]It actually has 15 ways. However, the datasets for two of them were corrupted and hence we only used 13.

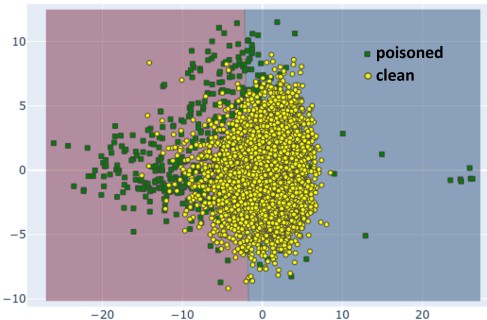

Figure 3: Projection and clustering results of TRACEBACK on clean-label attack. TRACEBACK does NOT have a clear separation between clean and poisoned samples.

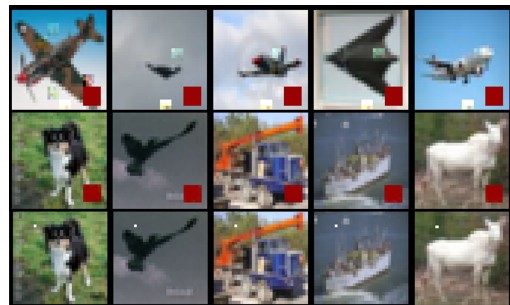

Figure 4: $D^3$ outperforms ABS in extracting poisons for CIFAR10's clean-label attack. The top row shows ground-truth triggers (red square). $D^3$'s extractions are accurate (2nd row), while ABS yields simpler triggers (white dot, 3rd row).

The other three datasets are poisoned by BadNet Gu et al. [2019], Trojnn Liu et al. [2018], the clean label attack Turner et al. [2019], dynamic backdoor attack Salem et al. [2020], and input-aware dynamic backdoor attack Nguyen and Tran [2020]. In total, we evaluate $D^3$ on 42 poisoned datasets. In comparison, TRACEBACK was evaluated on 5 datasets, which are included in ours as well, except for the Wenger Face dataset that we failed to gain access to from the authors.

Besides TRACEBACK, we further compare $D^3$ with Activation Clustering Chen et al. [2018], Spectral Signature Tran et al. [2018], and STRIP Gao et al. [2019a] that are the state-of-the-art poisoned sample detection approaches (Section B.2). In addition, we show that $D^3$ is superior to directly using triggers inverted by existing backdoor scanning technique ABS Liu et al. [2019] and FeatureRE Wang et al. [2022] (Section 3.2, we neglect comparison to NC since ABS usually outperforms NC).

In Section C.5, we demonstrate that even if not all the captured samples contain the ground-truth triggers, $D^3$ still manages to detox the training set with high precision and recall. We conduct ablation study to demonstrate $D^3$ remains effective under impact of poisoning rate (Section C.1), the validation set size (Section C.2), the number of captured poisoned samples (Section C.3), and model architecture (Section C.6). We also show the importance of data augmentation in Section D. Finally, we verify our technique is still effective against an adaptive attack (Section E). We discuss the limitations of our work in Section G.

**Running Time** Our evaluation is conducted on a server equipped with two Intel Xeon CPUs and an NVIDIA RTX A6000 GPU with 49140MiB memory. The average run time cost for scanning a dataset is 12.92 minutes, where the poison extraction step costs 9.26 minutes.

## B.1 Comparison with TRACEBACK (Cont.)

### B.1.1 Evaluation on TrojAI Datasets

**Experiment Setup.** For each poison type (e.g., triangle patch and Gotham filer), we randomly select 3 datasets with the type of poison using the random seed 0. There are 1000 samples in each class. The poisoning rate is 20%, meaning there are 200 additional poisoned images in the target class. For each dataset, we randomly select 10 poisoned samples as the test samples acquired in the wild.

Following the default setting of TrojAI competitions, we assume 200 clean samples from the victim class and the target class, respectively, as the hold-out validation set. Detoxing performance is hence evaluated on the remaining 800 clean samples in the target class together with the 200 poisoned samples. We study the effects of validation set size and number of poisoned samples in Section C.1 and Section C.2.

We extract poisons from the 10 poisoned test samples, stamp them on the victim class clean samples in the validation set, and perform standard image augmentation to generate the training set for the poisoned data classifier. The classifier uses SVM and is trained on the logits of augmented samples and clean target class samples (by the trojaned model). Finally, we apply the trained SVM to measure detoxing performance.

Table 3 shows the precision and recall of our method on the 39 TrojAI datasets. As the table shows, $D^3$ achieves nearly 100% precision and recall on most poisoned datasets, regardless of the poison

| Model ID | Attack Type | Trigger Type | Our Method | | TRACEBACK | |
|---|---|---|---|---|---|---|
| | | | Prec. (%) | Recall (%) | Prec. (%) | Recall (%) |
| 688 | polygon | 3 | 100.0 | 100.0 | 100.0 | 100.0 |
| 884 | polygon | 3 | 99.8 | 100.0 | 100.0 | 100.0 |
| 538 | polygon | 3 | **100.0** | **100.0** | 20.0 | 100.0 |
| 313 | polygon | 4 | **99.8** | **100.0** | 20.0 | 100.0 |
| 526 | polygon | 4 | **100.0** | **100.0** | 20.0 | 100.0 |
| 494 | polygon | 4 | **100.0** | **100.0** | 100.0 | 87.5 |
| 585 | polygon | 5 | **100.0** | **100.0** | 20.0 | 100.0 |
| 882 | polygon | 5 | **100.0** | **71.3** | 1.9 | 13.8 |
| 43 | polygon | 5 | **89.8** | **92.3** | 0.0 | 0.0 |
| 385 | polygon | 6 | **89.3** | 66.8 | 0.0 | 0.0 |
| 827 | polygon | 6 | 99.5 | 100.0 | 95.2 | 100.0 |
| 727 | polygon | 6 | 100.0 | 100.0 | 100.0 | 97.5 |
| 999 | polygon | 7 | **87.7** | **84.0** | 0.0 | 0.0 |
| 351 | polygon | 7 | 60.5 | 68.5 | 0.0 | 0.0 |
| 643 | polygon | 7 | 99.8 | 99.8 | 100.0 | 100.0 |
| 1005 | polygon | 8 | **98.5** | **100.0** | 100.0 | 6.3 |
| 386 | polygon | 8 | 99.8 | 100.0 | 100.0 | 97.5 |
| 183 | polygon | 8 | **97.3** | **100.0** | 0.0 | 0.0 |
| 598 | polygon | 9 | **96.4** | **99.8** | 20.8 | 20.0 |
| 893 | polygon | 9 | **99.0** | **100.0** | 0.0 | 0.0 |
| 533 | polygon | 9 | **100.0** | **100.0** | 20.0 | 100.0 |
| 135 | polygon | 10 | **97.0** | **90.0** | 20.0 | 100.0 |
| 1058 | polygon | 10 | 92.0 | 40.0 | 0.0 | 0.0 |
| 1060 | polygon | 10 | **99.8** | **100.0** | 90.9 | 100.0 |
| 876 | polygon | 11 | 82.4 | 90.0 | 20.0 | 100.0 |
| 79 | polygon | 11 | **99.5** | **97.5** | 1.8 | 5.0 |
| 710 | Polygon | 11 | 99.3 | 33.3 | 0.0 | 0.0 |
| 760 | polygon | 12 | **95.0** | **95.8** | 65.8 | 98.8 |
| 933 | polygon | 12 | **100.0** | **99.5** | 20.0 | 100.0 |
| 354 | polygon | 12 | **98.8** | **100.0** | 0.0 | 0.0 |
| 729 | Instagram | Kelvin | **93.9** | **99.8** | 20.0 | 100.0 |
| 398 | Instagram | Kelvin | **99.8** | **100.0** | 20.0 | 100.0 |
| 215 | Instagram | Kelvin | 100.0 | 100.0 | 100.0 | 100.0 |
| 696 | Instagram | Gotham | 100.0 | 100.0 | 100.0 | 100.0 |
| 903 | Instagram | Gotham | **99.5** | **100.0** | 20.0 | 100.0 |
| 977 | Instagram | Gotham | **100.0** | **100.0** | 0.0 | 0.0 |
| 476 | Instagram | Lomo | **100.0** | **100.0** | 0.0 | 0.0 |
| 259 | Instagram | Lomo | **100.0** | **100.0** | 0.0 | 0.0 |
| 1004 | Instagram | Lomo | **100.0** | **100.0** | 0.0 | 0.0 |

Table 3: Performance of $D^3$ and TRACEBACK on TrojAI dataset. The **bold** highlights the results of $D^3$ that are considerably better than TRACEBACK.

| Model ID | $D^3$ | | AC | | SS | | STRIP | |
|---|---|---|---|---|---|---|---|---|
| | Prec. (%) | Recall (%) | Prec.(%) | Recall(%) | Prec.(%) | Recall(%) | Prec.(%) | Recall(%) |
| 710 | 99.3 | 33.3 | 52.0 | 89.5 | 46.0 | 69.0 | 0.0 | 0.0 |
| 385 | 89.3 | 66.8 | 96.0 | 36.0 | 45.3 | 68.0 | 100.0 | 74.0 |
| 1058 | 92.0 | 40.0 | 33.9 | 75.0 | 44.3 | 66.5 | 99.2 | 30.3 |
| 43 | **89.8** | **92.3** | 15.2 | 34.5 | 4.0 | 6.0 | 0.0 | 0.0 |
| 827 | **99.5** | **100.0** | 0.0 | 0.0 | 0.0 | 0.0 | 0.0 | 0.0 |
| 760 | **95.0** | **95.8** | 58.5 | 100.0 | 54.7 | 82.0 | 0.0 | 0.0 |
| 538 | **100.0** | **100.0** | 100.0 | 100.0 | 62.7 | 94.0 | 16.7 | 0.8 |
| 903 | 99.5 | 100.0 | 100.0 | 100.0 | 66.7 | 100.0 | 0.0 | 0.0 |
| 882 | **100.0** | **71.3** | 45.0 | 83.5 | 44.7 | 67.0 | 0.0 | 0.0 |
| 585 | **100.0** | **100.0** | 100.0 | 99.5 | 65.7 | 98.5 | 98.6 | 18.0 |
| Clean-label | **100.0** | **94.0** | 7.6 | 38.0 | 72.3 | 43.4 | 72.0 | 35.0 |
| Dynamic | **100.0** | **99.3** | 99.9 | 94.9 | 0.0 | 0.0 | 88.9 | 6.8 |
| Input-aware | **96.7** | **90.1** | 7.5 | 6.7 | 39.7 | 2.4 | 0.0 | 0.0 |

Table 4: Comparison with Activation Clustering (AC), Spectral Signature (SS) and STRIP. $D^3$ outperforms them all.

| Model ID | Attack | Trigger | PR =0.2* | | PR=0.1 | | PR=0.15 | | VS=50 | | VS=20 | |
|---|---|---|---|---|---|---|---|---|---|---|---|---|
| | | | Prec. (%) | Recall(%) | Prec. (%) | Recall (%) | Prec. (%) | Recall(%) | Prec. (%) | Recall(%) | Prec. (%) | Recall (%) |
| 688 | polygon | 3 | **100** | **100** | **100** | **100** | **100** | **100** | **100** | **100** | **100** | **100** |
| 313 | polygon | 4 | **99.8** | **100** | 98.8 | **100** | 99.4 | **100** | 99.8 | **100** | 99.5 | **100** |
| 585 | polygon | 5 | **100** | **100** | 100 | 100 | 100 | 100 | 99.8 | 99.8 | 99.7 | 99.5 |
| 385 | polygon | 6 | **89.3** | 66.8 | 65.6 | 76.3 | 78.9 | 75.0 | 89.6 | 68.8 | **90.0** | 65.5 |
| 999 | polygon | 7 | **87.7** | 84.0 | 59.5 | 86.3 | 74.0 | 83.8 | 91.9 | 88.3 | **93.1** | 84.8 |
| 1005 | polygon | 8 | **100** | **99.5** | 100 | 98.8 | 100 | 99.4 | 98.8 | **100** | 95.9 | 99.5 |
| 598 | polygon | 9 | **96.4** | 99.8 | 84.2 | 100 | 91.4 | 100 | 93.2 | 100 | **92.0** | 100 |
| 135 | polygon | 10 | **93.5** | 97.5 | 74.3 | 97.5 | 85.0 | 95.6 | 97.5 | 88.5 | **97.0** | 90.0 |
| 876 | polygon | 11 | 82.4 | 90.0 | 49.0 | 92.5 | 65.5 | 91.3 | 87.0 | 83.5 | 87.1 | 84.3 |
| 760 | polygon | 12 | **95.0** | 95.8 | 79.8 | 98.8 | 88.8 | 98.8 | 96.9 | 86.9 | **95.1** | 91.5 |
| 729 | Instagram | Kelvin | 93.9 | 99.8 | 75.5 | 100 | 86.0 | 100 | 84.4 | 65.0 | 61.1 | 31.0 |
| 696 | Instagram | Kelvin | **100** | **100** | **100** | **100** | **100** | **100** | **100** | 97.0 | **100** | 82.5 |
| 476 | Instagram | Lomo | **100** | **100** | **100** | **100** | **100** | **100** | **100** | **100** | **100** | **100** |

Table 5: $D^3$'s Performance is hardly affected by validation set size and poisoning rate. * Indicates the default setting. PR=Poisoning Rate, VS=Validation Set Size

types. However, TRACEBACK is not stable on the TrojAI datasets because its clustering step cannot effectively distinguish poisoned and clean samples (as illustrated in Figure 3). Note that in many cases the precision and recall of TRACEBACK are 0. Further inspection shows that the method produces completely wrong separation at the clustering stage (e.g., considering a poisoned cluster as benign). In some cases, it predicts that the entire training set to be benign and hence has 0 true positives and 0 false positives (in poisoned sample prediction).

### B.2 Comparison with Other Related Works

We further compare $D^3$ with 3 closely related works, namely, Activation Clustering (AC), Spectral Signature (SS), and STRIP. Similar to TRACEBACK, AC divides the training set into two clusters by applying K-means on the dimensionality-reduced activations of the last hidden layer. After that, they predict the poisoned cluster(s) by retraining the model with one cluster and testing on the other, which is computationally expensive, or simply comparing the sizes of the two clusters. SS identifies poisoned samples by examining whether it has a special property, called *spectral signature*, which is commonly present in most poisons. STRIP detects poisoned inputs by intentionally perturbing them and observing the entropy of the predicted labels.

In this experiment, we use another random seed 43530870 to eliminate the bias of seed and select 10 TrojAI poisoned datasets. In addition, we include the clean label attack, dynamic backdoor attack, and input-aware backdoor attack.

Table 4 shows the results. Observe that $D^3$ outperforms all the three competitors. AC has perfect precision and recall for a few cases, on which $D^3$ also performs well. However, it does not have matching performance for the other cases, *e.g.*, the dataset poisoned by the clean label attack and the input-aware backdoor attack. This is because AC is based on clustering, which has similar limitations to TRACEBACK. SS does not work well because its detection of outliers of the property may not be effective. STRIP relies on the distribution of the validation set to compute an effective entropy threshold. When using the same setting as $D^3$, e.g., 200 validation samples in the victim class and the target class, the distribution of validation set is skewed from that of the training set, resulting in poor performance. Furthermore, the trigger can be corrupted when STRIP superimposes images on the poisoned training samples. Thus, the prediction entropy of those samples is also large.

# C   Ablation Study

In this section, we show the ablation study of poisoning rate, the size of validation set, the number of known poisoned samples, hyper-parameters, and model architecture.

## C.1   Impact of Poisoning Rate

In Table 5, we pick the first model under each trigger type from Table 3 and evaluate $D^3$'s precision and recall under poisoning rate 0.1 and 0.15 compared to the default setting poisoning rate 0.2. There are about half models that maintain high precision and recall (above 95%) regardless of the poisoning rate. On the other models, the precision drops a little with the decrease of poisoning rate. This is acceptable because there are fewer positive (poisoned) samples in a dataset with a smaller poisoning rate. Note that the recall remains high in a smaller poisoning rate setting, suggesting our technique has few false negatives and captures as many suspicious samples as possible.

## C.2   Impact of Validation Set Size

We use the same models as in Section C.1 to evaluate the impact of the validation set size. As Table 5 shows, it is note-worthy that $D^3$ maintains a high performance on most models, except for model #729, even when only given extremely small validation set (20 samples in victim class and 20 in target class). This enables individual users, who usually have no access to a large validation set, to effectively examine the training data of the model they purchased from vendors or downloaded online. For some models, e.g., model #999, the precision even increases when the validation set is smaller. This may be due to the trigger in this model having simple features such that a small set is adequate to train the classifier. More validation samples introduce noisy features (e.g., features specific to the validation samples) and distract the classifier from the trigger. The reason for the precision decrease on model #729 when given a small validation set is also intuitive, as the augmented dataset built on it is not large enough to train the classifier. As a simple verification for this hypothesis, the trigger for model #999 is only applied to local area whereas the trigger for model #729 is globally applied.

## C.3   Impact of Known Poisoned Samples

| | Model ID | 688 | 313 | 585 | 385 | 999 | 1005 | 598 |
|---|---|---|---|---|---|---|---|---|
| #p=5 | Prec. (%) | 100.0 | 99.8 | 100.0 | 93.6 | 85.1 | 100.0 | 98.3 |
| | Recall (%) | 100.0 | 100.0 | 100.0 | 62.0 | 50.0 | 100.0 | 98.5 |
| #p=3 | Prec. (%) | 100.0 | 99.8 | 100.0 | 90.6 | 82.5 | 100.0 | 97.1 |
| | Recall (%) | 100.0 | 100.0 | 100.0 | 65.3 | 29.5 | 98.5 | 99.0 |

Table 6: Ablation study on the number of poisoned samples.

In Table 6, we show that $D^3$ achieves high precision and recall even with fewer captured samples. Take half models from Table 5 as examples, the results suggest 3 poisoned samples are adequate for $D^3$ to achieve higher than 97.0% precision and recall for 5 out of 7 models.

## C.4   Impact of Hyper Parameters

We show $D^3$ is robust to hyper parameters in Table 7 by evaluating the dynamic backdoor attack setting. The default value for $\alpha$ is 10, while other values, 1 and 100, also achieve similar performance.

## C.5   Impact of the Captured Samples without Ground-truth Trigger

A possible scenario is that the captured misclassified inputs do not all contain the ground-truth. Instead, partial misclassification cases are caused by natural backdoors Tao et al. [2022c] or the deficiency of the model itself. Thus, we investigate how the ratio of #samples with ground-truth triggers (denoted as #G) over #all captured samples (denoted as #A) impacts $D^3$'s performance. Specifically, compared to the default setting #G=#A=10, we examine two additional combinations, #G=5, #A=10 and #G=5, #A=20, against dynamic backdoor attack. As shown in Table 7, the ratio of #G/#A has no significant impact on $D^3$'s performance, which makes our assumption more realistic.

| $\alpha$ | Prec. (%) | Recall (%) | (#G/#A) | Prec.(%) | Recall(%) |
|---|---|---|---|---|---|
| 1 | 100.0 | 99.1 | (10/10) | 100.0 | 99.3 |
| 10 | 100.0 | 99.3 | (5/10) | 100.0 | 99.1 |
| 100 | 100.0 | 99.3 | (5/20) | 100.0 | 99.4 |

Table 7: Ablation study on hyper parameter $\alpha$ and the ratio of samples with ground-truth trigger over all captured misclassified samples.

| Model ID | Architecture | Prec.(%) | Recall (%) |
|---|---|---|---|
| 122 | ShuffleNetV2 | 95.0 | 100 |
| 7 | WideResNet50 | 99.2 | 97.0 |
| 781 | GoogleNet | 97.8 | 100 |
| 143 | VGG11 | 95.7 | 88.5 |
| 555 | SqueezeNetV1_0 | 97.8 | 100 |
| 364 | densenet201 | 99.8 | 100 |
| 280 | resnet18 | 94.2 | 94.0 |

Table 8: $D^3$ is architecture-agnostic.

## C.6 Impact of Model Architecture

To show that our technique is architecture-agnostic, we randomly pick 7 models with different architectures using seed 0 from the TrojAI models poisoned with the quadrilateral patch as the poison. Table 8 shows the results. The precisions on the 7 models are all above 94% and half of them are above 97%. $D^3$ also achieves 100% recall on 4 out of the 7 models.

## D Necessity of Data Augmentation

We justify the necessity of data augmentation (i.e., creating more poisoned samples by stamping extracted poisons) by comparing our classifier with another SVM classifier trained on 10 known poisoned samples and 200 clean validation samples. Figure 5 shows the predictions of the SVM classifier on the full training set of TrojAI model #563, where it predicts the samples in the pink area as clean and the samples in the blue area as poisoned. The darker color the area has, the more confident the SVM is about its predictions. From the visualization, we can tell that the SVM has about 50% accuracy and does not do well on the poisoned samples (the data points in green) since it only "saw" limited poisoned samples. On the other hand, trained on augmented data, our SVM achieves 96.3% precision and 100.0% recall. This illustrates the importance of data augmentation.

## E Adaptive Attack

We assume the attacker adopts the following logits-hidden adaptive approach: besides the model to poison, the attacker also has a reference model that has the same structure and is trained on the same dataset (except not having any poison); when training a model using the poisoned dataset, the attacker wants to achieve a high ASR and a high accuracy on clean samples, as well as a minimal $L2$ distance between the logits of the poisoned model and the reference model given poisoned images. Formally, the adaptive loss is as follows.

$$L_{adaptive} = CE(M(x), y) + CE(M(x \oplus t), y_t)$$
$$+ \alpha || M(x \oplus t), M_c(x \oplus t) ||_2^2, \tag{9}$$

where $CE(\cdot)$ denotes the cross-entropy loss, $M$ the poisoned model, $M_c$ the clean reference model and $\alpha$ the adaptive criterion. It tends to mitigate the difference between the benign logits $M_c(x \oplus t)$ and the poisoned logits $M(x \oplus t)$ that $D^3$ uses to recognize malicious inputs. We use the BadNet attack and the CIFAR10 dataset setting for the experiment. We observe that a high ASR and a small logits distance are contradictory goals. Finally, we add the weight of $1e - 4$ for the adaptive criterion and achieve 95.41% ASR and 86.71% accuracy after 80 epochs of training. For the adaptively poisoned model and dataset, $D^3$ still achieves 99.5% precision and 99% recall, implying $D^3$ is robust to the adaptive attack.

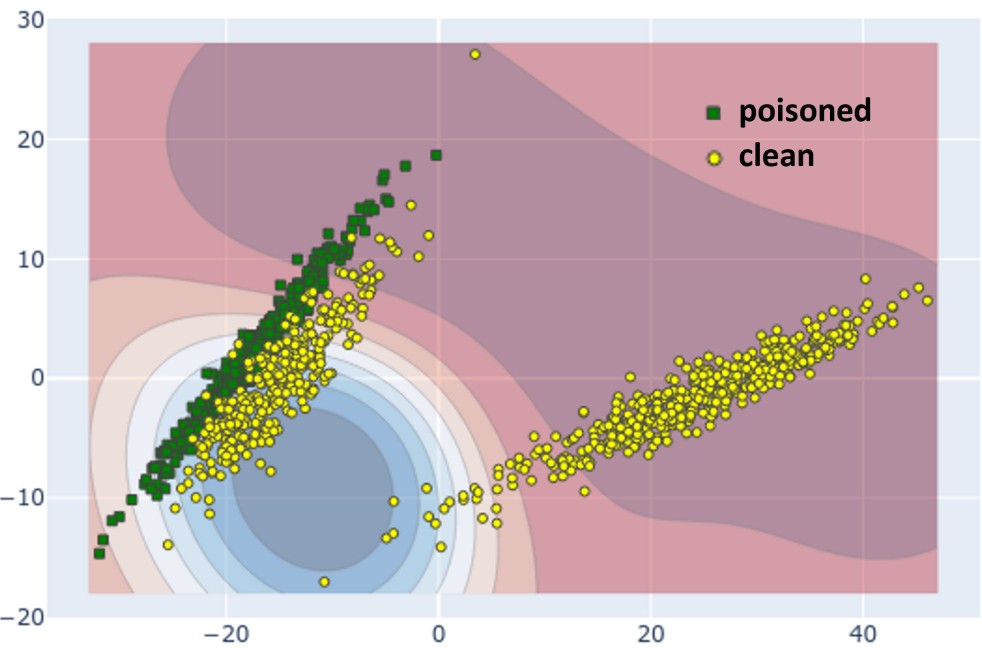

Figure 5: The SVM prediction probability of poisoning and clean samples only given captured samples.

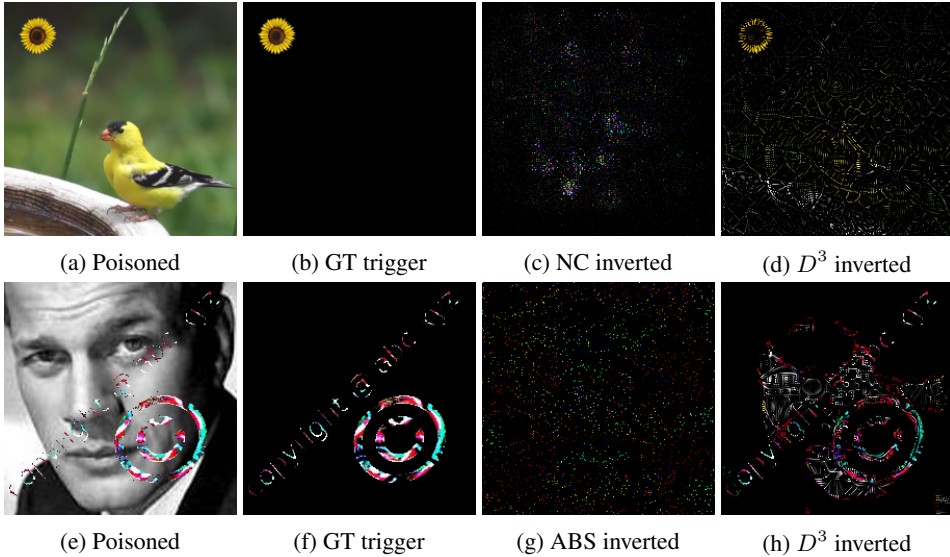

| (a) Poisoned | (b) GT trigger | (c) NC inverted | (d) $D^3$ inverted |

| (e) Poisoned | (f) GT trigger | (g) ABS inverted | (h) $D^3$ inverted |

Figure 6: Examples of extracted poisons. $D^3$ can better extract both the patch poison and the pervasive poison than state-of-the-art backdoor scanners ABS and NC.

## F   More Examples

Figure 6 shows some examples of extracted poisons. Specifically, Figure (d) shows our technique precisely extracts the yellow flower patch at the top left of an ImageNet sample in (a) poisoned by BadNet Gu et al. [2019], in comparison to the trigger inverted by a popular backdoor scanner NC Wang et al. [2019] in (c). Ours has more resemblance to the ground truth in (b). Figure (h) illustrates $D^3$ is also able to extract a watermark type of poison from a VGGFace sample poisoned by TrojNN Liu et al. [2018]. Similarly, it has more resemblance to the ground truth in (f), compared to the trigger inverted by ABS Liu et al. [2019], another popular backdoor scanner. More examples can be found in Section 3.2.

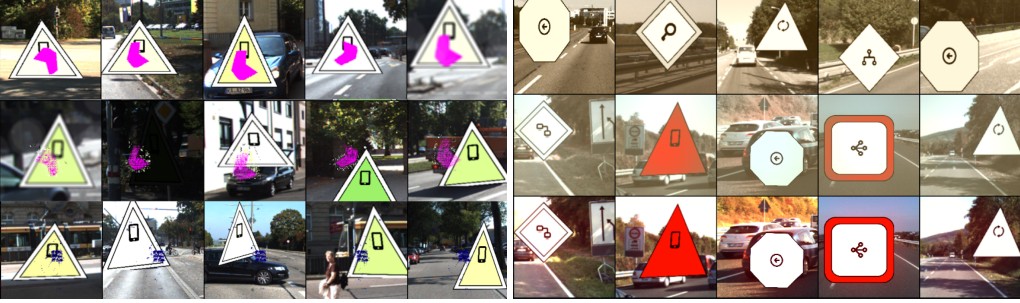

(a) Triggers extracted by $D^3$ and inverted by ABS for TrojAI model #52 poisoned by a patch.

(b) Triggers extracted by $D^3$ and inverted by ABS for TrojAI model #719 poisoned by a filter.

Figure 7: The first row shows the original poisoned samples with the ground-truth triggers (pink polygon for model #52 and filter for model #719.). The second row shows samples stamped with the poison extracted by $D^3$. The third row shows the samples stamped with the trigger inverted by ABS. Observe that $D^3$ can extract poison that resembles the ground-truth.

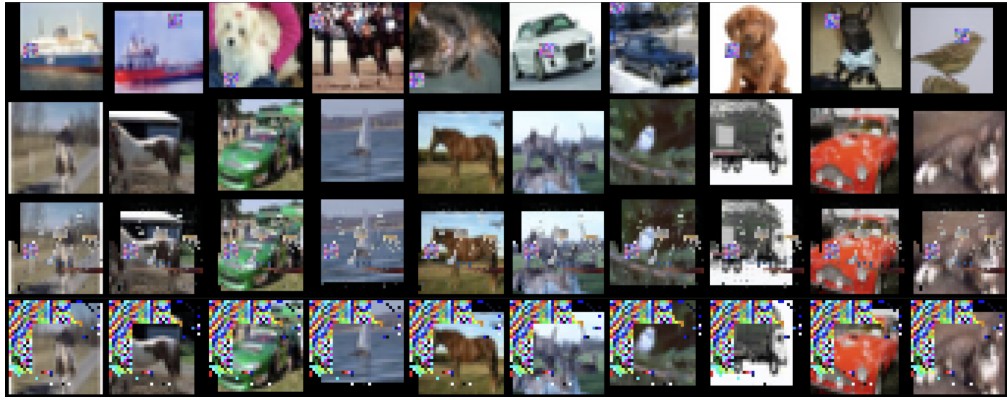

(a) The $D^3$-extracted poisons (row #3) in dynamic backdoor attack share similar patterns as the ground-truth triggers (row #1, the color square placed in various positions), whereas ABS-generated poisons (row #4) are different and can not trigger misclassifications.

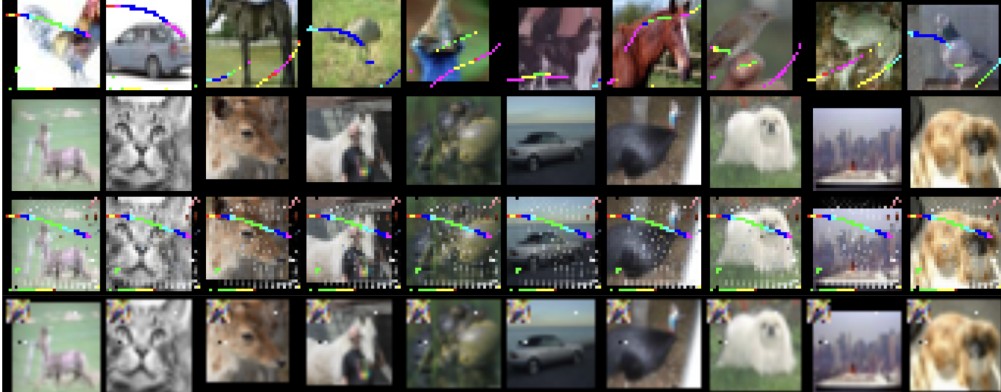

(b) The $D^3$-extracted poisons (row #3) in input-aware backdoor attack share similar patterns as the ground-truth triggers (row #1, the colorful strip), whereas ABS-generated poisons (row #4, the pixel square on top left) are different and can not trigger misclassifications.

Figure 8: More examples from the CIFAR10 dataset suggest that $D^3$-extracted poisons resemble the ground-truth triggers more compared to the poisons reverse-engineered by ABS. The first row in each subfigure shows the captured poisoned samples with ground-truth triggers; the second rows present the clean validation samples; in the third rows, we show the clean validation samples stamped by $D^3$-extracted poisons; the validation samples stamped by ABS-extracted poisons are shown in the last rows.

| Model ID | $D^3$ | | ABS | | FeatureRE | |
|---|---|---|---|---|---|---|
| | Prec.(%) | Recall (%) | Prec.(%) | Recall(%) | Prec.(%) | Recall(%) |
| 1058 | 92.0 | 40.0 | 80.3 | 39.8 | 0.0 | 0.0 |
| 585 | **100.0** | **100.0** | 100.0 | 99.3 | 94.6 | 96.5 |
| 999 | 87.7 | 84.0 | 100.0 | 20.3 | 100.0 | 74.8 |
| 688 | **100.0** | **100.0** | 0.0 | 0.0 | 100.0 | 1.0 |
| 385 | 89.3 | 66.8 | 100.0 | 40.0 | 100.0 | 94.5 |
| 727 | **100.0** | **100.0** | 0.0 | 0.0 | 93.2 | 100.0 |
| 876 | 82.4 | 90.0 | 86.3 | 96.0 | 97.9 | 71.5 |
| 827 | **99.5** | **100.0** | 0.0 | 0.0 | 100.0 | 4.8 |
| 933 | **100.0** | **99.5** | 100.0 | 93.3 | 0.0 | 0.0 |
| 598 | **96.4** | **99.8** | 100.0 | 71.5 | 0.0 | 0.0 |
| Clean-label | **100.0** | **94.0** | 12.8 | 47.0 | 77.7 | 99.6 |
| Dynamic | **100.0** | **99.3** | 0.0 | 0.0 | 100.0 | 14.1 |
| Input-aware | **96.7** | **90.1** | 0.0 | 0.0 | 82.2 | 6.4 |

Table 9: Comparison of poison extracted by $D^3$ with by black-box reverse engineering tools ABS and FeatureRE. $D^3$ has overall better performance, indicating the knowledge of poisoned test examples plays an important role in generating effective triggers.

Next, we consider the all-to-one dynamic backdoor attack and input-aware attack, where the target classes are both the airplane and all the other classes are victims. Figure 8 shows the $D^3$-extracted poisons in the third rows resemble the ground-truth triggers more compared to the poisons generated by ABS, shown in the fourth rows. Note that even for these attacks whose triggers are specific to the poisoned samples, $D^3$ can still find a universal substitute poison that causes misclassification when stamped on a random clean image.

## G   Discussion

Unlike other existing backdoor defenses, both the baseline TRACEBACK and our proposed method, $D^3$, incorporate the assumption of a few poisoned test samples captured in the wild. It's essential to emphasize that this does not imply TRACEBACK and $D^3$ are less effective than other related works. Instead, this distinction arises from our focus on a novel problem: the forensic setting.

In an environment where increasingly sophisticated attacks can circumvent current defense strategies, our attention shifts to the post-mortem scenario. We are concerned with understanding how, once the attacker has breached the defenses, we can learn from the incident. By analyzing captured poisoned test samples, we aim to trace back the training samples that facilitated the attack.

The similar threat model has precedent in traditional software security and is proven to be of vital importance in system protection. Aligning with this mindset, $D^3$ contribute to providing a robust complement to existing defenses, particularly when they fail to detect complex, stealthy poisoning attacks.

While our proposed method $D^3$ demonstrates significant improvements over existing baselines, we must acknowledge certain limitations in our approach.

**Scope of Application:**   The applicability of $D^3$ is constrained primarily to the Computer Vision (CV) domain and detoxify image sets. This limitation arises from our employment of a pre-trained StyleGAN to generate $x$ from the optimizable noise $z$.

**Optimization Strategy:**   In our design of $D^3$, we created separate optimization formulas for patch triggers and pervasive triggers, the two prominent classes into which the literature on stealthy poisoning can be categorized. Though it may be viewed as a limitation, this division aligns with common practice. For example, ABS classifies triggers into two categories: 'simple' and 'complex.' It then analyzes the input patterns associated with simple triggers, while employing artificial brain stimulation techniques to address complex triggers.

**Detection Capability:**   $D^3$ is capable of detecting only those poisoned samples that correspond to the same attack types found in the captured test samples. It's crucial to clarify, however, that this does not mean the triggers from the captured samples must match those in the training data. Since $D^3$ focuses on the feature level (i.e., logits), it is equipped to handle scenarios where the triggers in captured samples differ from the training data. Examples include dynamic backdoor attacks, input-aware backdoor attacks, and clean-label attacks.

