# OpenReview forum: "$D^3$: Detoxing Deep Learning Dataset"
_NeurIPS.cc/2023/Workshop/BUGS — NeurIPS 2023 BUGS Poster_

### Official Review · Reviewer_Bacv · 2023-10-22
**Review for Submission #12**

**Rating:** 4
**Confidence:** 5

**Review:**

## Pros

* This work introduces a multi-stage poison dataset detpxification pipeline.
* The poison extraction technique is rather novel, by separating the poison trigger and the clean images given some actual poison samples.
* The experimental results show that the proposed method indeed surpass the baselines in some cases. The authors also provide several ablation studies and an adaptive attack in the Appendix.

## Cons & Questions

- The **assumption** of the defender is too strong -- you require access to 1) some actual poisoned samples (and the underlying target class); 2) additional clean samples (and the underlying victim class); 3) the poisoned training set used to train the model. These requirements significantly diminish the practicality of the defense. Meanwhile, it also leads to much unfairness when comparing with other baseline methods (e.g., SS, STRIP and AC).
- What about all-to-all attack? Can $D^3$ effectively mitigate it?
- The **principles** behind the methodology are not well explained. For example, in Eq (7), why would the intuition of "enforcing style similarity" help recover invisible noise-like trigger pattern (e.g., WaNet)? Also, for patch-like poison, why "encourage pixels undergo either no changes or maximum changes", considering that actual patch triggers are not necessarily with 255 pixel magnitudes?
- In line 162, you are saying your **motivation** of recovering poison is because you have to complement the insufficient poison samples. What if you just *assign larger weights* to (gradients update of) the "very few poisoned samples" you have when training such a classifier? Such an additional study would be more persuasive. In addition, I suggest you emphasize this motivation somewhere earlier in the paper.
- In line 169, you use $\tilde p$, a small perturbation, in addition to the recovered poison trigger pattern $p$. Does this imply that your poison extraction method cannot recover small-perturbation triggers (e.g., WaNet)?
- In line 171, why are you using "logit values" but not representation-level information to distinguish clean and poison samples? I suspect logit values convey information that are too little and naive to well distinguish clean and poison samples. Some additional evidence for such a choice would be necessary.
- In Sec 3.2, why compare with black-box backdoor scanner but not white-box ones? This is not fair (other than the unfairness w.r.t. posioned test samples you have mentioned), considering you assumed access to the poisoned model internal details (white-box level access). How would white-box trigger reverse engineering method (e.g. Neural Cleanse [1]) performs when compared with yours?
- In Line 201, why trian a classifier in the feature space for FeatureRE, which is different from what you did for both $D^3$ and ABS?
- Consider comparing with more advanced poison cleansers (e.g., [2] and [3]).
- Overall, I feel the method pipeline is too complicated, in the sense of #hyperparameters (e.g. the choice of $\gamma$ is not well studied) and the lack of interpretability of each component. Thus, its practicality needs more ellaboration.

[1] Wang, Bolun, Yuanshun Yao, Shawn Shan, Huiying Li, Bimal Viswanath, Haitao Zheng, and Ben Y. Zhao. "Neural cleanse: Identifying and mitigating backdoor attacks in neural networks." In 2019 IEEE Symposium on Security and Privacy (SP), pp. 707-723. IEEE, 2019.

[2] Qi, Xiangyu, Tinghao Xie, Jiachen T. Wang, Tong Wu, Saeed Mahloujifar, and Prateek Mittal. "Towards a proactive {ML} approach for detecting backdoor poison samples." In *32nd USENIX Security Symposium (USENIX Security 23)*, pp. 1685-1702. 2023.

[3] Tang, Di, XiaoFeng Wang, Haixu Tang, and Kehuan Zhang. "Demon in the variant: Statistical analysis of {DNNs} for robust backdoor contamination detection." In *30th USENIX Security Symposium (USENIX Security 21)*, pp. 1541-1558. 2021.

---

### Official Review · Reviewer_RHyA · 2023-10-27

**Rating:** 6
**Confidence:** 4

**Review:**

This paper falls within the scope of poison forensics.
Given some captured poison samples, the paper proposes to extract the trigger pattern from these poison samples. Then, these extracted triggers can be used to train a classifier to separate clean and poison samples.

The approach makes a lot of sense, while the weakness seems to be the prior assumption on the form of the trigger.
For example, in the paper, it is assumed to be:
(1) Patch-like (equation-2)
(2) or a linear transformation like (equation-6)

However, backdoor triggers may take a lot of different forms, beyond these two.
For example, rotation-based [1].
Also, I wonder whether the approach is still effective when training time and test time triggers are asymmetric [2]. Say, if the test time trigger captured is different from the training-time one, I expect the classifier can fall shot of detecting poison samples in the training dataset.


[1] Wu, T., Wang, T., Sehwag, V., Mahloujifar, S. and Mittal, P., 2022, November. Just rotate it: Deploying backdoor attacks via rotation transformation. In Proceedings of the 15th ACM Workshop on Artificial Intelligence and Security (pp. 91-102).

[2] Qi, X., Xie, T., Li, Y., Mahloujifar, S. and Mittal, P., 2022, September. Revisiting the assumption of latent separability for backdoor defenses. In The eleventh international conference on learning representations.

---

### Decision · Program_Chairs · 2023-10-28

**Decision:**

Accept (Poster)

**Comment:**

This paper presents a novel approach to poison forensics, focusing on trigger pattern extraction and subsequent classification. Specifically, the authors introduced a multi-stage poison dataset detoxification pipeline and an innovative poison extraction technique. The authors should further discuss the assumptions and practicality of the defenses, as suggested by the reviewers.